# HCV poly U/UC sequence–induced inflammation leads to metabolic disorders in vulvar lichen sclerosis

Qing Cong[1],*, Xiao Guo[1],*, Shengwei Zhang[1],* , Jinhui Wang[1], Yi Zhu[2], Lili Wang[1], Guangxing Lu[1] , Yufeng Zhang[1], Wei Fu[1], Liying Zhou[1], Shuaikang Wang[1], Cenxi Liu[1], Jia Song[3], Chaoyong Yang[3,4], Chi Luo[2], Ting Ni[1], Long Sui[1] , He Huang[1] , Jin Li[1]

Vulvar lichen sclerosis (VLS) is a dermatologic disorder that affects women worldwide. Women with VLS have white, atrophic papules on the vulva. They suffer from life-long intense pruritus. Corticosteroids are the first-line of treatments and the most effective medicines for VLS. Although VLS has been speculated as an autoimmune disease for a long time, its pathogenesis and the molecular mechanism is largely unknown. We performed a comprehensive multi-omics analysis of paired samples from VLS patients as well as healthy donors. From the RNA-seq analysis, we found that VLS is correlated to abnormal antivirus response because of the presence of Hepatitis C Virus poly U/UC sequences. Lipidomic and metabolomic analysis revealed that inflammation-induced metabolic disorders of fatty acids and glutathione were likely the reasons for pruritus, atrophy, and pigment loss in the vulva. Thus, the present study provides an initial interpretation of the pathogenesis and molecular mechanism of VLS and suggests that metabolic disorders that affect the vulva may serve as therapeutic targets for VLS.

## Introduction

Vulvar lichen sclerosis (VLS) is a rare, chronic, progressive dermatologic disorder. The incidence of VLS worldwide has not been reported yet, although an epidemiology study in the Netherlands revealed that 7.4–14.6 per 100,000 women of all ages have VLS (Bleeker et al, 2016). Most VLS patients are aware of the disease because of the intense pruritus that often interferes with sleep (Val & Almeida, 2005). White, atrophic papules in the patients may coalesce into plaques on the vulva, which can also deteriorate if

patients receive no proper treatment (Powell & Wojnarowska, 1999). Histologically, VLS biopsies present hyperkeratosis, significant thinning with loss of the normal rete ridge pattern, and plugging of follicular infundibulate (Tallon, 2011). However, the diagnosis of VLS can be made solely based on clinical features. Histological examination is recommended if there are atypical features or diagnostic uncertainty (Lewis et al, 2018). VLS is a risk factor for various kinds of malignant squamous cell neoplasia of the vulva, including invasive squamous cell carcinoma and vulvar intraepithelial neoplasia (Bleeker et al, 2016; Micheletti et al, 2016).

Corticosteroids are the first-line treatments and the most effective medicines for VLS currently (Lee & Fischer, 2018; Lewis et al, 2018). Most VLS patients receive a life-long corticosteroid treatment, even though the long-term utilization of corticosteroid can cause multiple side effects such as cutaneous atrophy, severe redness, and soreness. Recurrence of VLS may occur if the patients stop using corticosteroids spontaneously. Alternatively, photodynamic therapy was reported to remit VLS by eliciting cell death and could be applied to patients who are unresponsive to corticosteroid treatment (Belotto et al, 2017). As far as we know, no targeted therapy has been identified for VLS.

Although VLS is a serious skin disease affecting women in an annoying, embarrassing, and long-term way across the globe, its pathogenesis and molecular characteristics are still undefined. Previous studies have suggested a correlation between inflammation or microRNA and VLS, but the lack of comprehensive understanding of its mechanism prevents the development of precision therapy. VLS has been speculated as an autoimmune disease (Tran et al, 2019). This hypothesis was derived from several clinical observations: (1) VLS is frequently observed in women with Turner syndrome (Chakhtoura et al, 2014); (2) many VLS patients also developed various autoimmune diseases (Kreuter et al, 2013; Song et al, 2018); (3) VLS patients may also have a higher frequency of certain HLA antigens,

[1]Obstetrics and Gynecology Hospital, State Key Laboratory of Genetic Engineering, Institute of Metabolism and Integrative Biology and School of Life Sciences, Fudan University, Shanghai, China   [2]Institute of Environmental Medicine and Department of Hepatobiliary and Pancreatic Surgery, The First Affiliated Hospital, Zhejiang University School of Medicine, Hangzhou, China   [3]Institute of Molecular Medicine, Renji Hospital, School of Medicine, Shanghai Jiao Tong University, Shanghai, China   [4]The Ministry of Education Key Laboratory of Spectrochemical Analysis and Instrumentation, State Key Laboratory for Physical Chemistry of Solid Surfaces, Key Laboratory for Chemical Biology of Fujian Province, College of Chemistry and Chemical Engineering, Xiamen University, Xiamen, China

Correspondence: suilong@fudan.edu.cn; he_huang@fudan.edu.cn; li_jin_lifescience@fudan.edu.cn
*Qing Cong, Xiao Guo, and Shengwei Zhang contributed equally to this work
Jin Li is senior author

 Life Science Alliance

although few immunological consequences have been observed (Sideri et al, 1988). Microarray-based expression analysis revealed up-regulation of Th1, microRNA-155, and other related transcripts using VLS and healthy donor biopsies (Terlou et al, 2012; Ren et al, 2018). However, person-to-person genotypic variation makes it difficult to identify molecular markers without using normal tissues from the same patient, and the PCR-based measurement of transcriptome in microarray assay limited the data interpretation and assumption validation.

In this article, we performed multi-omics analyses on VLS-paired samples from the same patients as well as the normal ones from healthy donors. RNA-seq analysis revealed that VLS patients are suffering from abnormally high inflammation with impressive antivirus features. In-depth analysis of their transcriptomes identified a remarkable enrichment of hepatitis C virus (HCV) poly uridine/uridine-cytidine (U/UC) sequences, which might contribute to the onset of VLS. Furthermore, the multi-omics results indicated various inflammation-induced metabolic disorders in VLS samples, which might serve as therapeutic targets for precision medicine.

## Results

### Abnormal activation of antivirus response in VLS

To systematically identify the pathological causes of VLS, we carefully recruited VLS patients without exposure to any treatment (see the information listed in Table S1) and collected the diagnosed VLS skin tissue (i.e., VLS group) and adjacent tissue (i.e., control group) from the same VLS patient to minimized genotypic variation. Considering the same genotype of the paired VLS biopsies, the transcriptome alterations might be indicative of dominant effects that trigger VLS.

Thus, we first performed RNA-seq analysis of the VLS paired samples. The expression matrix is presented in Table S2. Differential expression analysis identified that 371 and 331 genes were significantly up-regulated or down-regulated in the VLS group compared with the control group, respectively (Fig 1A). This result suggests the remarkable difference of transcriptome between VLS samples and control samples. Of these 281 significantly highly expressed genes, the results of functional analysis with Gene Set Enrichment Analysis (Subramanian et al, 2005) surprisingly identified a set of hallmark genes associated with inflammation (Fig 1B). We then further confirmed that the up-regulated genes were extensively linked to innate immunity, particularly pertaining to RNA virus infection responses in the VLS group (Fig 1C) using Gene Ontology analysis. Specifically, canonical antivirus genes, including interferons (*IFN*), members of the 2–5 A synthetase family, chemokines, DEAD-box proteins, and *TLRs*, exhibited increased expression in the VLS group compared with the non-symptomatic control (Fig 1D). These transcriptomic results indicated a potential correlation between responses to virus infection and VLS pathogenesis.

### Enrichment of noncoding virus sequences in VLS samples

It has long been speculated that either viruses or NCVSs are responsible for the onset of various autoimmune diseases (Nexø et al,

2016). According to our findings from the VLS transcriptome, it is reasonable to hypothesize that VLS could also be induced by viruses or NCVS. To test this hypothesis, we aligned and compared both the VLS and healthy donor raw RNA-seq reads to the reference genome of all viruses with human hosts in GenBank individually. The normalized abundance of transcripts was comparable between VLS patients and healthy donors for most virus types (Fig S1), whereas the HCV presented a remarkably higher normalized abundance in VLS patients (Fig 2A). We then examined the accumulated HCV sequences in VLS patients, which revealed a specific enrichment of poly U/UC sequences in the 3' NTR of the HCV genome (Fig 2B).

The poly U/UC sequence in the HCV 3' NTR is a well-known NCVS. It has been widely recognized that HCV poly U/UC sequences alone can induce robust antivirus responses (Saito et al, 2008; Schnell et al, 2012). On the contrary to the highly accumulated poly U/UC sequence, the absence of other portions of the HCV genome indicated that the enrichment of NCVS was a genetic polymorphism rather than virus infection in VLS patients. The poly U/UC sequences can integrate into the human genome via a similar approach as retrovirus sequences (Kvaratskhelia et al, 2014; Tang et al, 2020).

To investigate the transcriptional dysregulation of poly U/UC sequences in VLS patients, we performed de novo assembly to identify their locations on the human genome. It was interesting that the poly U/UC sequences in healthy donors were located in either introns or lowly expressed exons, but the poly U/UC sequences in VLS patients were located in highly expressed exons (Fig 2C). We presented one example about the different localization of the poly U/UC sequences in healthy donors and VLS patients: the poly U/UC sequences were inserted in the *HSD17B12* intron in one healthy donor, whereas these sequences were fused to the last exon of the highly expressed ferritin light chain (*FTL*) in one VLS patient (Fig 2D). All of these findings suggested that the genetic polymorphism of NCVS sequences in the human genome might contribute to the abnormal inflammation of VLS.

### Inflammation-associated metabolomic alterations in VLS

Multiple studies suggested that abnormal inflammation, induced by viral infection or NCVS, can lead to metabolic changes in various cell types (Steckelberg et al, 2018; Ketter & Randall, 2019). Applying our transcriptomic data to the REACTOME metabolic network database, we detected that genes related to fatty acid and glutathione (GSH) metabolism were dramatically changed in the VLS group (Fig 3A and B). Thus, we applied lipidomic and metabolomic approaches to investigate the alterations of lipids and hydrophilic metabolites of VLS-paired samples. The lipidomic analysis revealed a reduction in long-chain fatty acids (LCFAs) and very-long-chain fatty acid (VLCFAs) in VLS samples (Fig 3C and Table S3), which indicated that the fatty acid elongation pathway was perturbed in VLS patients as we observed from transcriptomic results. Both LCFAs and VLCFAs are well-known to serve as important components of the epidermal barrier. Thus, a deficiency of these fatty acids could contribute to skin damage as well as itchiness in VLS patients.

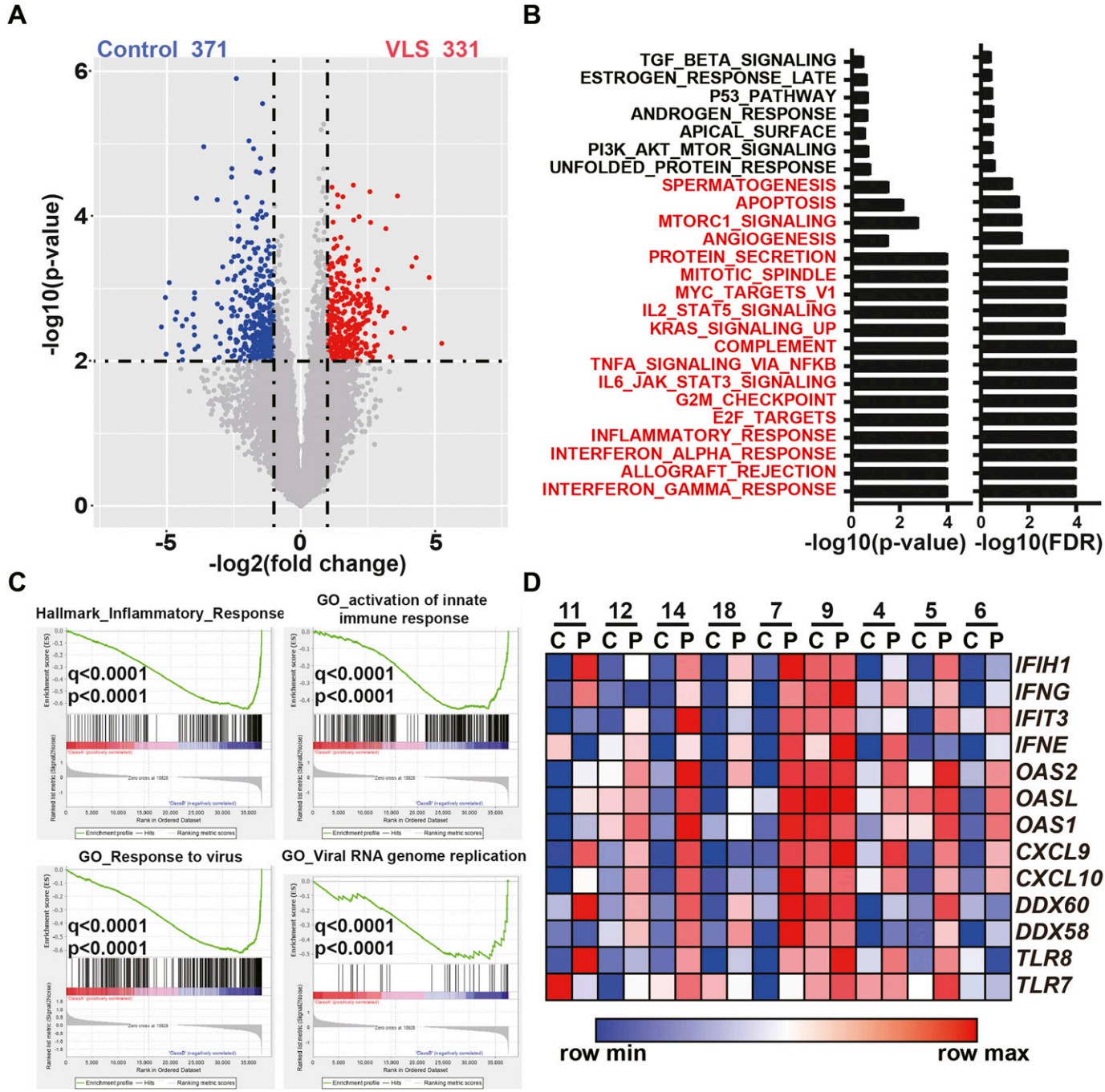

**Figure 1. Up-regulation of antivirus genes in vulvar lichen sclerosis (VLS) samples.**
**(A)** RNA-seq revealed 371 up-regulated and 331 down-regulated genes in VLS samples compared to controls. **(B)** Gene Set Enrichment Analysis identified genes enriched in inflammation response in VLS samples. **(C)** Gene Set Enrichment Analysis identified up-regulated antivirus genes in VLS samples. **(D)** Heat map showing expression of antivirus genes of each pair of samples.

Although we observed fewer hydrophilic metabolite dysregulations in the VLS group than we acquired from our lipidomic results (Fig S2), a heat map of the top 70 differentially detected metabolites still indicated an increase in the ratio between GSH and its oxidized product glutathione disulfide (GSSG) (Fig 3D and E and Table S4),

which might be affected by glutathione-S-transferase (GST). GSH usually forms glutathione-S-conjugates to function as a detoxicant and antioxidant (Lee et al, 1997); thus, the up-regulated GSH/GSSG ratio indicated that the abnormal inflammation in VLS patients could potentially be linked to this xenobiotic metabolic disorder.

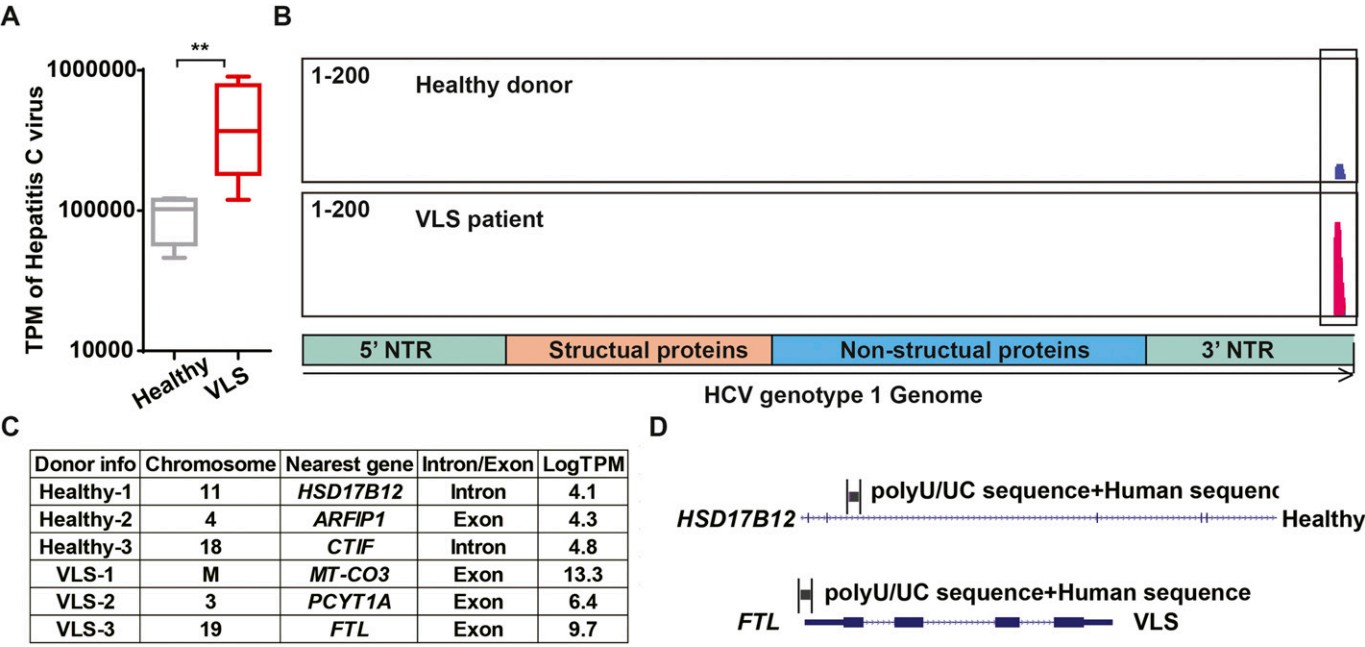

**Figure 2. Enrichment of hepatitis C virus (HCV) poly U/UC sequences in vulvar lichen sclerosis (VLS) samples.**
**(A)** Transcript per million of HCV sequences in VLS patient samples and healthy donor samples. **P < 0.01 of U-test. **(B)** An example of HCV sequences in a VLS patient sample and a healthy donor sample. **(C)** The integration site of HCV poly U/UC sequences in VLS patients and healthy donors. **(D)** An example of HCV poly U/UC sequences in a VLS patient and a healthy donor.

### Defects of GSH metabolism leads to impairment of melanocytes

Melanocyte is one of the major affected cell types in VLS, and we observed a dramatic loss of melanocytes or melanin in VLS biopsies as expected (Fig 3F). In contrast, we did not identify dramatic changes of melanocyte-related gene expression in RNA-seq analysis (Fig S3). During melanin biosynthesis, melanocytes generate large amounts of reactive oxygen species, which are then detoxified by cellular antioxidant systems such as GSH. Without sufficient GSH metabolism to eliminate the accumulated reactive oxygen species, melanocytes could hardly maintain their normal function, likely leading to the loss of pigments in VLS patients. To investigate whether melanocyte viability and identity is dependent on GSH metabolism, we inhibited the activities of GST in the melanocytic cell line B16 by pan GST inhibitors GSTO-IN-2 (Chang et al, 2006) or ethacrynic acid (Ploemen et al, 1990). The inhibitors, as well as glutathione itself, decreased pigment production in B16 cells upon stimulation with forskolin and tyrosine (Fig 3G and H). These results highlighted the significance of maintaining metabolic homeostasis for melanocyte functionality, which could also contribute to the VLS treatment.

## Discussion

VLS is a rebarbative disease that harms both the physical and psychological health of women. These features make the fundamental investigation of VLS urgently needed. Although VLS has been proposed as an inflammatory disease for years, there are still three major unaddressed issues: (1) What kind of inflammation occurs in the VLS? (2) What causes inflammation in VLS patients? (3) How does the inflammation induce skin damage and lead to VLS?

To begin to answer these and other questions germane to VLS, we conducted a multi-omics study to understand the molecular characteristics associated with VLS. In this study, we applied both RNA-seq analysis and mass spectrometry (MS)–based metabolomic/lipidomic analyses to study the molecular characteristics of VLS. Using the RNA-seq data, we found the up-regulation of genes related to innate immunity and antivirus response in VLS samples, and further used targeted virus genome analyses to reveal an enrichment of HCV 3′ poly-U/UC sequences in VLS-affected tissues.

The connection between the presence of a virus and the onset of autoimmune diseases has been proposed for decades. For example, a few studies suggested that the onset of Type 1 Diabetes is correlated to exposure to various kinds of viruses in childhood (Marguerat et al, 2004; Levet et al, 2019). But these studies are mostly observational and this hypothesis has not been proven experimentally. In this study, we identified the poly U/UC 3′ sequences of HCV as a potential pathogenic factor that is responsible for VLS. Further studies on experimental models are required to validate the pathogenic effects of HCV 3′ poly U/UC sequence. In addition, the fact that only the vulva of these patients is affected suggests the involvement of environmental factors (such as life style and microbiota) in the onset of VLS.

We further explored the consequences of hyperinflammation using the transcriptomic results and associated some transcriptional changes to metabolic dysregulations in VLS patients. It has

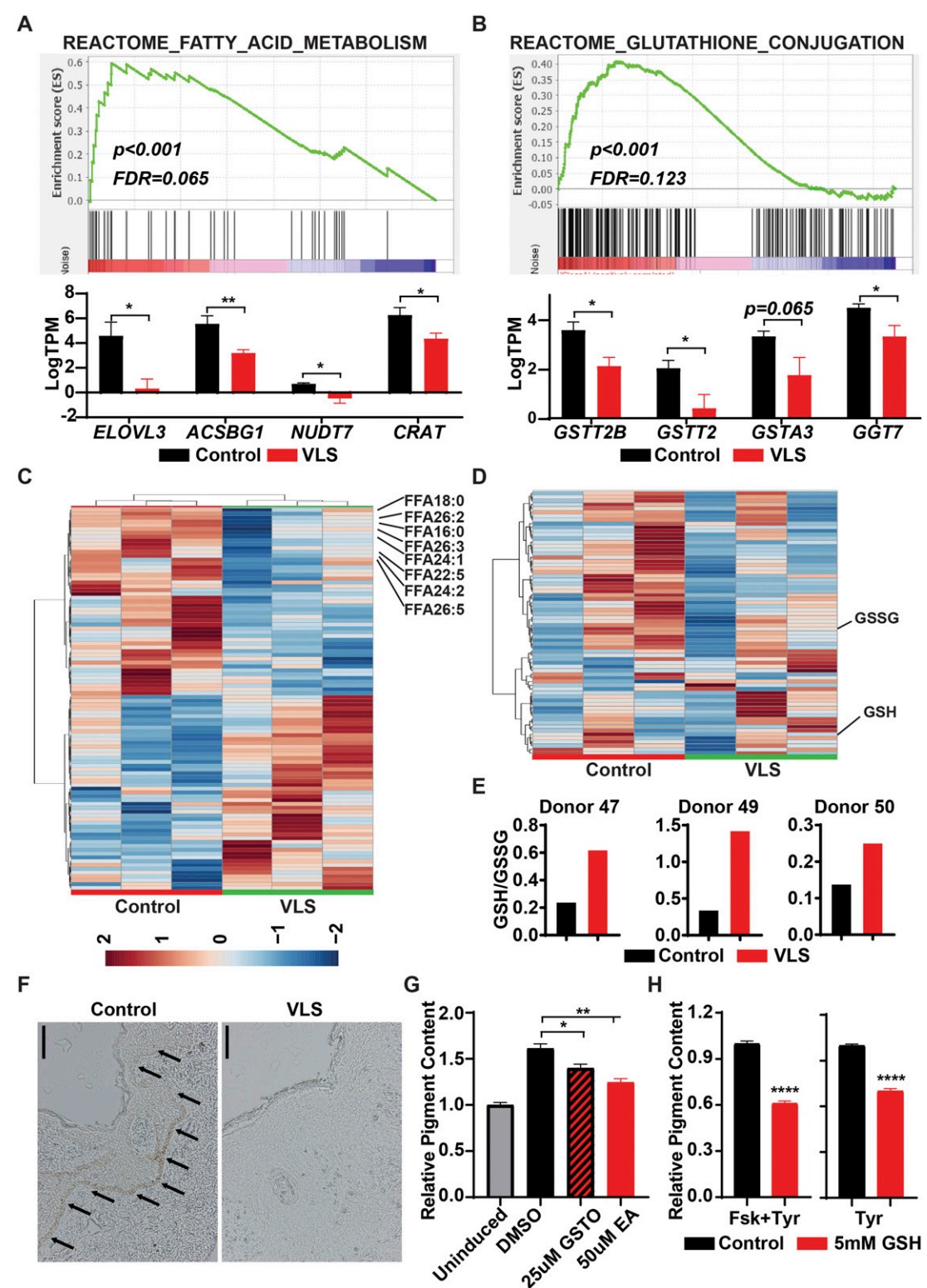

**Figure 3. Metabolic disorders in vulvar lichen sclerosis (VLS) samples.**
**(A)** Gene Set Enrichment Analysis identified down-regulation of fatty acid elongation genes in VLS samples. *P < 0.05; **P < 0.01. **(B)** Gene Set Enrichment Analysis identified down-regulation of glutathione-s-transferase genes in VLS samples. *P < 0.05. **(C)** Heat map showing lipidomic changes in VLS samples. **(D)** Heat map showing metabolic changes in VLS samples. **(E)** The ratio of GSH/GSSG in paired samples from VLS patients. **(F)** Histology of paired samples from VLS patients. Scale bar, 100 μm. **(G)** Changes of intracellular pigment with GST inhibitors treatment. *P < 0.05; **P < 0.01. **(H)** Changes of intracellular pigment with GSH treatment. ****P < 0.0001.

been shown that NCVS can affect the metabolism of cells or tissues. Lipidomic and metabolomic analyses on paired samples indicated the dysregulation of VLCFA/LCFA and GSH metabolism in VLS-affected tissues. VLCFAs and LCFAs are major components of the skin barrier (Uchida, 2011; Cui et al, 2016) which can relieve the itchiness induced by the abnormal inflammation of the skin (Kaczmarski et al, 2013; Søyland et al, 1993, 1994). Thus, decreased levels of VLCFA and LCFA could be one of the reasons for the skin damage and itchiness in VLS patients. Another metabolite, glutathione, which was more highly accumulated in the VLS group is also critical for skin health via maintaining the homeostasis and function of melanocytes (Kondo et al, 2016; Sonthalia et al, 2018). The impairment of melanocyte function and cell identity using GST inhibitors and glutathione indicated that therapies regulating the metabolism of glutathione could be a potential therapy for VLS.

In summary, we performed a multi-omics analysis on VLS patient samples to understand their molecular characteristics. This study identified potential pathogenic factors and their impacts on VLS, which could serve as therapeutic targets in the future. Similar studies may need to carry out to understand the pathogenesis of extragenital LS.

# Materials and Methods

## Sample collection

The current study was approved by the ethical committee of Fudan University Obstetrics and Gynecology Hospital. Informed consent was obtained from all subjects. All the experiments in the current study conformed to the principles described in the World Medical Association Declaration of Helsinki and the Department of Health and Human Services Belmont Report. The patient and healthy donor information is summarized in Table S1.

## Reagents

All of the chemicals for MS, including methanol (HPLC grade, 99.9%), water, HPLC-grade water, liquid chromatography (LC)–MS grade acetonitrile (HPLC grade, 99.95%), methyl tert-butyl ether (HPLC-grade, 99.9%), ammonium acetate (LC–MS grade), and ammonium hydroxide (LC–MS grade) were ordered from Fisher Chemical. The GST inhibitors GSTO-IN-2, ethacrynic acid, and GSH were ordered from MedChemExpress.

## RNA-seq and analysis

Total RNA was extracted from normal and pathological tissues with the TRIzol reagent (Invitrogen). RNAs were then reversely transcribed with oligo(dT) primers. RNA-seq libraries for expression analysis were constructed using KAPA RNA HyperPrep Kit KR1350 v1.16 according to the vendor's protocol and paired-end 2 × 150 bp reads were sequenced using the Illumina HiSeq platform. The data were aligned and quantified by HISAT2 (Kim et al, 2019). The raw data are deposited at GSE166620.

## Viral genome assembly and sequence alignment

RNA-seq data were first mapped to human genome hg19 using HISAT2 with default parameters. SAMtools (Li et al, 2009) was next used to extract unmapped RNA-seq reads, within which virus-related reads were located. These unmapped paired-end reads were then mapped against a concatenated virus genome (600 viruses with human as a host from National Center for Biotechnology Information, downloaded from https://www.ncbi.nlm.nih.gov/genomes/GenomesGroup.cgi?taxid=10239&host=human) on April 9, 2019, using HISAT2. We performed deduplication for the alignment results to remove potential PCR duplicates. Trinity (Grabherr et al, 2011) was used to de novo assemble the RNA-seq data to explore whether some viral RNAs existed. Finally, Salmon was used to estimate the transcript-level abundance of alignment results for each virus (Patro et al, 2017).

## Hydrophilic metabolite extraction

Tissue samples were homogenized at –20°C for 1.5 h. Methanol:water (*v:v*, 80:20) was prechilled at –80°C overnight, and 4 ml was added to the tissue sample homogenate. The homogenate was then incubated at –80°C for 20 min and decanted to a 15-ml centrifuge tube. The homogenate was centrifuged at 4°C at 4,000$g$ for 10 min, and the supernatant was then collected in another 15-ml centrifuge tube. 500 $\mu$l of prechilled 80% methanol was added to the 15-ml centrifuge tube which contained the tissue homogenate, and after 1 min of vortexing, the tissue homogenate was centrifuged at 4°C at 4,000$g$ for 10 min again. Approximately 500 $\mu$l of supernatant was added to the ~4 ml of supernatant in a new 15-ml centrifuge tube. The 4.5 ml supernatant was split into three portions (3 × 1.5-ml microcentrifuge tubes). The 80% methanol extracted metabolites were then dried using a SpeedVac (LABCONCO Refrigerated CentriVap Concentrator) and stored at –80°C before MS analysis.

## Tissue lipid extraction

A tissue sample was added to 200 $\mu$l of water and 500 $\mu$l of methanol and homogenized using the same approach as in the hydrophilic metabolite extraction above. The homogenate was supplemented with 500 $\mu$l more methanol and decanted into a clean glass centrifuge tube. 5 ml of methyl tert-butyl ether was then added to the glass centrifuge tube and vortexed for 1 min. The glass centrifuge tube containing the homogenate was rocked on a shaker for 1 h at room temperature. 1.25 ml of water was then added to the glass centrifuge tube followed by another minute of vortexing. The homogenate was centrifuged at 4°C at 1,000$g$ for 10 min and two-phase layers could be observed in the glass centrifuge tube. 4 ml of the top phase supernatant were collected and dried under a stream of nitrogen. The extracted lipid sample was stored at –80°C before MS analysis.

## Targeted metabolomic analysis

The metabolomic approach was adopted from a published method (Zhang et al, 2019). In general, samples were resuspended in 50 $\mu$l of water:acetonitrile (*v:v*, 50:50), and 5 $\mu$l was injected into a 6500 QTRAP triple-quadrupole MS (SCIEX) coupled to an HPLC system (Shimadzu). Metabolites were eluted via hydrophilic interaction

chromatography (HILIC) by using a 4.6-mm i.d. × 10 cm Amide XBridge column (Waters) with a flow rate of 400 $\mu$l/min using buffer A (20 mM ammonium hydroxide/20 mM ammonium acetate (pH 9.2) at a 95:5 ratio with water:acetonitrile) and buffer B (acetonitrile). Gradients were run from 85% buffer B to 42% buffer B at 0–5 min; from 42% buffer B to 0% buffer B at 5–16 min; 0% buffer B was held from 16–24 min; from 0% buffer B to 85% buffer B at 24–25 min; and 85% buffer B was held for 7 min. All ions were acquired by 306 selected reaction monitoring transitions in a positive and negative mode switching fashion. Electrospray ionization (ESI) voltage was +4,900 and −4,500 V in positive or negative mode, respectively.

### Lipidomic analysis

Lipid samples were resuspended in 100 $\mu$l of chloroform:methanol: water ($v$:$v$:$v$, 45:45:10), and 5 $\mu$l was injected into a 5500 plus QTRAP triple-quadrupole MS (SCIEX) coupled to HPLC system (Shimadzu). Lipids were eluted via HILIC by using a 2.1 mm i.d. × 10 cm BEH column (Waters) with a flow rate of 500 $\mu$l/min using buffer A (10 mM ammonium acetate (pH 8) at a 95:5 ratio with acetonitrile:water) and buffer B (10 mM ammonium acetate (pH 8) at a 50:50 ratio with acetonitrile:water). Gradients were run from 0.1% buffer B to 20% buffer B at 0–10 min; from 20% buffer B to 98% buffer B at 10–11 min; 98% buffer B was held from 11–13 min; from 98% buffer B to 0.1% buffer B at 13.0–13.1 min; and 0.1% buffer B was held for 3 min. All the ions were acquired by 1250 selected reaction monitoring transitions associated with their predicted retention time in a positive and negative mode switching fashion. ESI voltage was +5,500 and −4,500 V in positive or negative mode, respectively.

### Pigment content assay

Murine melanocytic cell line B16 was maintained in RIPM 1640 with 10% FBS and 1% P/S, and cultured in a 5% $CO_2$ humidified incubator at 37°C. Melanogenesis was induced by either 200 $\mu$M L-Tyrosine (Sangon), or 200 $\mu$M L-Tyrosine and 5 $\mu$M forskolin (Sigma-Aldrich). For treatment with GST inhibitors or reduced GSH, equal amounts of B16 were initially seeded in maintenance medium for attachment, followed by treatment with vehicle control or indicated compounds in induction medium for 5 d. Melanin was quantified as described previously (Luo et al, 2015). In brief, cells were lysed by boiling in 1 N NaOH for 10 min, followed by measuring the absorbance at 420 nm with a Varioskan Flash (Thermo Fisher Scientific) plate reader.

### Bioinformatic analysis

All $P$-values were calculated via $t$ test unless specifically indicated. All the metabolomic and lipidomic data were analyzed by MetaboAnalyst software (https://www.metaboanalyst.ca/).

## Data Availability

The raw RNA-seq data are deposited at gene expression omnibus with accession number GSE166620.

## Supplementary Information

## Acknowledgements

This work was supported by the Shanghai Sailing Program (20YF1402600), Natural Science Foundation (NSF) of China 92057115, and National Key R&D Program of China 2020YFA0803800 and 2019YFA0801900 to H Huang, 2018YFA0801300 to J Li, NSF of Shanghai 20ZR1470900 to Q Cong, and NSF of Zhejiang LR20H160004 to C Luo.

### Author Contributions

Q Cong: conceptualization, data curation, investigation, methodology, and writing—original draft, review, and editing.
X Guo: conceptualization, data curation, investigation, methodology, and writing—original draft, review, and editing.
S Zhang: conceptualization, data curation, software, investigation, visualization, methodology, and writing—original draft, review, and editing.
J Wang: software, investigation, methodology, and writing—original draft, review, and editing.
Y Zhu: investigation, visualization, methodology, and writing—original draft, review, and editing.
L Wang: data curation, investigation, methodology, and writing—original draft, review, and editing.
G Lu: data curation, visualization, and writing—original draft, review, and editing.
Y Zhang: data curation, visualization, and writing—original draft, review, and editing.
W Fu: data curation, investigation, and writing—original draft, review, and editing.
L Zhou: data curation, investigation, visualization, methodology, and writing—original draft, review, and editing.
S Wang: investigation, methodology, and writing—original draft, review, and editing.
C Liu: investigation, methodology, and writing—original draft, review, and editing.
J Song: investigation, methodology, and writing—original draft, review, and editing.
C Yang: conceptualization, data curation, and writing—original draft, review, and editing.
C Luo: conceptualization, supervision, investigation, and writing—original draft, review, and editing.
T Ni: supervision, investigation, methodology, and writing—original draft, review, and editing.
L Sui: conceptualization, supervision, investigation, methodology, and writing—original draft, review, and editing.
H Huang: data curation, software, supervision, funding acquisition, investigation, methodology, and writing—original draft, review, and editing.
J Li: conceptualization, data curation, supervision, funding acquisition, investigation, visualization, methodology, and writing—original draft, review, and editing.

## Conflict of Interest Statement

The authors declare that they have no conflict of interest.

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
