## [Reviewer comments · Life Science Alliance]

Life Science Alliance

HCV poly U/UC sequence induced inflammation leads to metabolic disorders in Vulvar Lichen Sclerosis

Qing Cong, Xiao Guo, Shengwei Zhang, Jinhui Wang, Yi Zhu, Lili Wang, Guangxing Lu, Yufeng Zhang, Wei Fu, Liying Zhou, Shuaikang Wang, Cenxi Liu, Jia Song, Chaoyong Yang, Chi Luo, Ting Ni, Long Sui, He Huang, and Jin Li

DOI: <https://doi.org/10.26508/lsa.202000906>

Corresponding author(s): *Jin Li, Fudan University*

Review Timeline:	Submission Date:	2020-09-14
	Editorial Decision:	2021-02-05
	Revision Received:	2021-02-14
	Editorial Decision:	2021-06-03
	Revision Received:	2021-06-08
	Accepted:	2021-06-09

Scientific Editor: *Eric Sawey, PhD*

Transaction Report:

February 5, 2021

Re: Life Science Alliance manuscript #LSA-2020-00906-T

Jin Li
Fudan University
CHINA

Dear Dr. Li,

Thank you for submitting your manuscript entitled "HCV poly U/UC sequences induced abnormal inflammation leads to metabolic disorders in Vulvar Lichen Sclerosis" to Life Science Alliance. The manuscript was assessed by expert reviewers, whose comments are appended to this letter.

We apologize for this extreme delay in getting back to you. We ran into significant mishaps finding reviewers for this study, and unfortunately could not retrieve comments from a reviewer despite multiple efforts. As you will note below, the VLS expert was quite enthusiastic about these findings, and based on their input we would like to invite you back to submit a revised version to us. Since, we were unable to retrieve an opinion from a multi-omic expert this time, we might seek advice from one when the revised manuscript comes in - as a final advice. It is LSA's policy to only allow for one round of review, and so the expert will only be asked to comment on the solidity of the data, and not to provide a full review.

Thank you for this interesting contribution to Life Science Alliance. We are looking forward to receiving your revised manuscript.

Sincerely,

Shachi Bhatt, Ph.D.
Executive Editor
Life Science Alliance
<https://www.lsjournal.org/>
Tweet @SciBhatt @LSAJournal

Interested in an editorial career? EMBO Solutions is hiring a Scientific Editor to join the international Life Science Alliance team. Find out more here -
https://www.embo.org/documents/jobs/Vacancy_Notice_Scientific_editor_LSA.pdf

- A letter addressing the reviewers' comments point by point.
- An editable version of the final text (.DOC or .DOCX) is needed for copyediting (no PDFs).
- High-resolution figure, supplementary figure and video files uploaded as individual files: See our detailed guidelines for preparing your production-ready images, <https://www.life-science-alliance.org/authors>
- Summary blurb (enter in submission system): A short text summarizing in a single sentence the study (max. 200 characters including spaces). This text is used in conjunction with the titles of papers, hence should be informative and complementary to the title and running title. It should describe the context and significance of the findings for a general readership; it should be written in the present tense and refer to the work in the third person. Author names should not be mentioned.

B. MANUSCRIPT ORGANIZATION AND FORMATTING:

Reviewer #1 (Comments to the Authors (Required)):

VLS has been thought to be an autoimmune disorder. However, its pathogenesis and molecular mechanisms are relatively unknown.

The aim of this study was to carry out a multi-omic analysis of paired samples from CLS patients and healthy donors. They also carried out RNA-seq analysis, mass spectrometry based metabolic studies and virus genome targeted analysis.

They did find that VLS correlated to the antiviral response due to the presence of abnormal HCV poly U/UC sequences. This increase in HCV poly/UC sequences may contribute to VLS.

RNA-seq analysis in these patients also showed them to have abnormally high inflammation with anti-virus features. This inflammation induced metabolic disorders in fatty acid and glutathione.

Gene analysis, showed an upregulation of 281 of 351 genes. However, I am uncertain that this indicates a potential correlation between responses to virus infection and VLS pathogenesis.

Their main points are strongly supported by the data presented.

The authors study looked at multiple factors to get a better understanding of the pathogenesis of VLS and hopefully allow for the development of novel treatments such as GST inhibitors. They do comment that in these patients the disease was localized to the vulva and hence there could be other factors such as the environment playing a part.

In the discussion section the authors mention (3rd paragraph) the association between autoimmune diseases and presence of a virus. They quote many studies on Diabetes and viral exposure. In fact there are very few studies on this. One would need to elaborate on these studies. Additionally, they mention environmental factors being possibly involved. I would like some comments on these factors. They have highlighted the need for multi-omics analysis for precision medicine which I believe may be correct but is based only on 13 patients.

As a clinician I believe that this is an important study which contributes further to our understanding of VLS.

A comment on extragenital LS and the need for carrying out a similar analysis would be useful to see whether we are dealing with the same pathogenesis.

My only other comment is that the English may require some minor changes.

Response letter

Reviewer #1 (Comments to the Authors (Required)):

VLS has been thought to be an autoimmune disorder. However, its pathogenesis and molecular mechanisms are relatively unknown.

The aim of this study was to carry out a multi-omic analysis of paired samples from CLS patients and healthy donors. The also carried out RNA-seq analysis, mass spectrometry based metabolic studies and virus genome targeted analysis.

They did find that VLS correlated to the antiviral response due to the presence of abnormal HCV poly U/UC sequences. This increase in HCV poly/UC sequences may contribute to VLS.

RNA-seq analysis in these patients also showed them to have abnormally high inflammation with anti-virus features. This inflammation induced metabolic disorders in fatty acid and glutathinine.

Gene analysis, showed an upregulation of 281 of 351 genes. However, I am uncertain that this indicates a potential correlation between responses to virus infection and VLS pathogenesis.

*We agree with the reviewer that the number of significantly changed genes does not correlate to the responses to virus infection. We presented these numbers in order to show that the transcriptome of VLS samples and the control samples is different merely. The correlation between responses to virus infection and VLS pathogenesis was examined using viral genome assembly experiment in later section. We will make this point clearer, by adding the following sentence “**This result suggests the remarkable difference of transcriptome between VLS samples and control samples**” to the manuscript.*

Their main points are strongly supported by the data presented.

The authors study looked at multiple factors to get a better understanding of the pathogenesis of VLS and hopefully allow for the development of novel treatments such as GST inhibitors. They do comment that is these patients the disease was localized to the vulva and hence there could be other factors such as the environment playing a part.

In the discussion section the authors mention(3rd paragraph) the association between autoimmune diseases and presence of a virus. They quote many studies on Diabetes and viral exposure. In fact there are very few studies on this. One would need to elaborate on these studies. Additionally, they mention environmental factors being possibly involved. I would like some comments on these factors.

We thank the reviewer for the positive comments, finding our results firmly support our conclusions. We appreciate the reviewer's suggestions about the 3rd paragraph of discussion section. As the reviewer correctly indicated, there are studies about diabetes and virus exposure but not many. Therefore, we revise that sentence as "For example, a few studies suggested the onset of Type 1 Diabetes (T1D) is correlated to the exposure to various kind of virus in childhood". We further elaborate this phenomenon by adding a sentence as "But these studies are mostly observational and this hypothesis has not been proven experimentally". We also discussed the potential environmental factors for VLS by revising the sentence as "In addition, the fact that only the vulva of these patients is affected suggests the involvement of environmental factors (such as life style, microbiota, etc) to the onset of VLS".

They have highlighted the need for multi-omics analysis for precision medicine which I believe may be correct but is based only on 13 patients.

We agree with the comments from the reviewer. Since the number of patient samples in our study was relatively low, we would like to delete this sentence to avoid over-claiming.

As a clinician I believe that this is an important study which contributes further to our understanding of VLS.

A comment on extragenital LS and the need for carrying out a similar analysis would be useful to see whether we are dealing with the same pathogenesis.

The reviewer brings up an important aspects of LS study. To address this point, we have added a sentence in the last paragraph as "Similar studies may need to carry out to understand the pathogenesis of extragenital LS".

My only other comment is that the English may require some minor changes.

This comment has been taken seriously. The revised manuscript has been professionally edited for English usage by a native English speaker with expertise in biomedicine.

June 3, 2021

RE: Life Science Alliance Manuscript #LSA-2020-00906-TR

Dr. Jin Li
Fudan University
220 Songhu Road
Shanghai 200000
China

Dear Dr. Li,

Thank you for submitting your revised manuscript entitled "HCV poly U/UC sequence induced inflammation leads to metabolic disorders in Vulvar Lichen Sclerosis". We would be happy to publish your paper in Life Science Alliance pending final revisions necessary to meet our formatting guidelines.

- please edit the 2nd sentence of the Abstract to read: Women with VLS exhibit white, atrophic papules on the vulva and experience intense pruritus in a prolonged way.
- please rename the datasets as supplementary tables - both in their titles and in their callouts in the manuscript text;
- LSA allows supplementary figures, but no EV Figures; please update your callouts for the Supplementary Figures in the manuscript Fig EV1A=Fig S1A; while supplementary figures use the system supplementary Fig S1;
- the Supplemental Figures should also have complete Figure Legends
- please rename "Declaration of interests" section to "Conflict of interest"
- please add a Data Availability statement, including the RNA-seq dataset
- scale bars are needed for figure 3F, please indicate scale bar size in Legend

A. FINAL FILES:

B. MANUSCRIPT ORGANIZATION AND FORMATTING:

Sincerely,

Eric Sawey, PhD
Executive Editor
Life Science Alliance

June 9, 2021

RE: Life Science Alliance Manuscript #LSA-2020-00906-TRR

Dr. Jin Li
Fudan University
220 Songhu Road
Shanghai 200000
China

Dear Dr. Li,

Thank you for submitting your Research Article entitled "HCV poly U/UC sequence induced inflammation leads to metabolic disorders in Vulvar Lichen Sclerosi". It is a pleasure to let you know that your manuscript is now accepted for publication in Life Science Alliance. Congratulations on this interesting work.

*****IMPORTANT:** If you will be unreachable at any time, please provide us with the email address of an alternate author. Failure to respond to routine queries may lead to unavoidable delays in publication.*******

DISTRIBUTION OF MATERIALS:

Again, congratulations on a very nice paper. I hope you found the review process to be constructive and are pleased with how the manuscript was handled editorially. We look forward to future exciting submissions from your lab.

Sincerely,
